# Ozone exposure disrupts insect sexual communication

Nan-Ji Jiang [1,2], Hetan Chang[1], Jerrit Weißflog[3], Franziska Eberl[4], Daniel Veit[4], Kerstin Weniger[1], Bill S. Hansson[1,2,5] & Markus Knaden [1,2,5] ✉

Insect sexual communication often relies upon sex pheromones. Most insect pheromones, however, contain carbon-carbon double bonds and potentially degrade by oxidation. Here, we show that frequently reported increased levels of Anthropocenic ozone can oxidize all described male-specific pheromones of *Drosophila melanogaster*, resulting in reduced amounts of pheromones such as cis-Vaccenyl Acetate and (*Z*)−7-Tricosene. At the same time female acceptance of ozone-exposed males is significantly delayed. Interestingly, groups of ozone-exposed males also exhibit significantly increased levels of male-male courtship behaviour. When repeating similar experiments with nine other drosophilid species, we observe pheromone degradation and/or disrupted sex recognition in eight of them. Our data suggest that Anthropocenic levels of ozone can extensively oxidize double bonds in a variety of insect pheromones, thereby leading to deviations in sexual recognition.

Finding and judging a suitable mate is pivotal for reproduction in many animals. In this context, most insects use sex pheromones to discriminate conspecifics from allospecifics and to identify the sex and mating status of a potential mate[1–3]. A particularly well-investigated pheromone is cis-Vaccenyl Acetate (cVA). This compound is produced by male *Drosophila melanogaster*, governs sex recognition, and the cVA amount present on a male has been shown to correlate with the male's attractiveness to a female[4–7]. During copulation, however, the male, in order to ensure its paternity, transfers cVA to the female and thereby reduces the female's attractiveness to other males[6,8,9]. cVA is thus attractive to females but repulsive to males. Many fly species of the genus *Drosophila* are known to produce male specific compounds that increase their attractiveness towards conspecific females, become transferred during copulation, and, hence, seem to fulfill similar pheromone-like roles like cVA in *Drosophila melanogaster*[10,11]. Although being chemically diverse, most of these potential pheromones share one specific feature—they contain carbon double bonds.

During the Anthropocene, insects communicating with such unsaturated pheromones are facing a potential challenge: the oxidization of double bonds by increased levels of oxidant pollutants like ozone[12]. Pheromone systems have evolved in pre-industrial times with tropospheric ozone values as low as 10 ppb[13]. However, due to the continuous emission of nitrogen oxides (NOx), volatile organic compounds (VOCs) and the climatic change, the ozone level has already increased to a global yearly average of 40 ppb[14]. Local extreme ozone events have been reported for industrial and urban areas of e.g., Mexico, Bangladesh, Morocco, and China[15–18]. Ozone reached in Mexico a concentration of up to 210 ppb (that lasted an hour) and the highest mean value measured over 10 h exceeded 170 ppb[15]. Although ozone values vary a lot during the whole year, the yearly measured average for e.g., north-eastern China has increased from 45 ppb in 2003 to 62 ppb in 2015[16] and reached a daily maximum averaged for 8 h (MDA8) of 140 ppb observed in March 2020[19].

Here we show, that even short-term exposure to ozone levels of 100 ppb results in the degradation of many drosophilid pheromones and reduces e.g., the attractiveness of males to females in 7 of 10 tested species. Interestingly, ozone-exposure dramatically increases male-male courtship behavior, probably due to a lack of sex discrimination when male pheromones become degraded.

[1]Department of Evolutionary Neuroethology, Max Planck Institute for Chemical Ecology, Hans-Knöll-Straße 8, D-07745 Jena, Germany. [2]Next Generation Insect Chemical Ecology, Max Planck Centre, Max Planck Institute for Chemical Ecology, Hans-Knöll-Straße 8, D-07745 Jena, Germany. [3]Mass Spectrometry/ Proteomics Research Group, Max Planck Institute for Chemical Ecology, Hans-Knöll-Straße 8, D-07745 Jena, Germany. [4]Max Planck Institute for Chemical Ecology, Hans-Knöll Straße 8, D-07745 Jena, Germany. [5]These authors contributed equally: Bill S. Hansson, Markus Knaden. ✉e-mail: mknaden@ice.mpg.de

## Results and discussion

We first investigated, whether the amount of cVA of *D. melanogaster* (CS) males becomes affected by exposure to ozone. Indeed, when comparing with control flies exposed to ambient air (with 4.5 ± 0.5 ppb ozone) only, we found reduced amounts of cVA on flies that were exposed to 100 ppb ozone for 2 h (Fig. 1, for a schematic of the ozone setup see Fig. S1) and increased amounts of heptanal (Fig. S2), a potential breakdown product of cVA oxidation. Interestingly, many other pheromone compounds like (*Z*)−7-Tricosene (7-T) and (*Z*)−7-Pentasene (7-P)[20], which are known to be involved in reproductive behavior, were decreased after ozone exposure, as well (Fig. 1c).

We next asked whether these changes of the males' chemical profiles would affect their attractiveness to female flies. We therefore exposed male flies either for 30 min to ozone ranging from 50 to 200 ppb or, as a control, to ambient air and afterwards tested their courtship behavior and mating success with non-exposed females in a no-choice mating assay (Fig. 2a). Ozone-exposed males did not differ from control males regarding their courtship latency (i.e., the time until they started to court the female; Fig. 2a), and courtship percentage (i.e., the percentage of males that courted females, Fig. S3), showing that male courtship motivation does not seem to be affected by previous exposure to ozone. At the same time, although most males finally mated within the 10 min observation (Fig. S3) ozone-exposed males exhibited a longer mating latency than control males (i.e., they needed more time to become accepted by the female; Fig. 2a). Our results indicate that ozone-exposed males were less attractive to the courted females, which corresponds well with the aforementioned reduction of pheromones upon ozone exposure. We did not observe any effect of ozone when flies were only exposed for 15 min, while the mating latency of males again was increased after ozone exposure for two hours or after exposure to higher levels of ozone (150 ppb and 200 ppb) (Fig. S4). Interestingly, after being exposed to ozone for 2 h, males did not recover their original chemical profile and their attractiveness to females after 1 day but exhibited normal pheromone levels and attractiveness after 5 days (Fig. S5).

As mentioned before, male-specific pheromones often do not only function as aphrodisiacs for females but also help males to discriminate sexes[6,7]. We, therefore, hypothesized that ozone exposure of male flies, with the subsequent reduction of their male-specific pheromones, would impede sex discrimination. When we exposed groups of males to 100 ppb of ozone, the results, however, exceeded our expectations. After a brief period (17.72 ± 1.83 min, *N* = 10) of exposure, males started to court each other intensively and to exhibit chaining behavior (supp. Movie S1 and S2), i.e., formed a long chain of courting males that was first described for males carrying a *fruitless* mutation, i.e., a mutation in the *fruitless* gene that changes the males' mating preferences[21]. When quantifying such male-male courtship behavior of pairs of ozone-exposed males (i.e., 30 min at 100 ppb) in a no-choice assay (Fig. 2b), we indeed found a higher number of trials that resulted in male courtship as compared to air-exposed control males. Again, the effect of ozone could be increased by either prolonging the exposure time or increasing the level of ozone (Fig. S6). As both males were exposed to ozone in these experiments, we wondered, whether the observed increased male-male courtship was only due to the degraded male-specific pheromones, or was in addition affected by a potential mal-function of sensory neurons, responsible for their detection, due to oxidative stress. Single sensillum recordings (SSR) from antennal trichoid sensilla at1, known to house an olfactory sensory neuron that detects cVA[7], revealed similar dose-response curves in ozone-exposed flies as in control flies (Fig. 2c). Furthermore, we confronted an intact male with a decapitated male and exposed each of them either to ozone or ambient air before the encounter. We found that whether or not the intact male would exhibit courtship behavior was not influenced by its own exposure to ozone but only by the exposure of the decapitated male (Fig. 2d). To sum up, ozone exposure does not impede the function of pheromone responsive neurons or other levels of signal processing, but rather induces male-male courtship via the degradation of pheromones. We next asked whether ozone exposure impedes sex-discrimination completely, and analyzed the courting preference of non-ozone-exposed males towards a decapitated male and female in a choice assay. Before the test, the decapitated flies were exposed to either ozone or control air. While the males preferentially courted the females in the control experiment, ozone exposure of the decapitated flies resulted in equal courting of female and male flies (Fig. 2e). Interestingly, when only the decapitated male was exposed to ozone before the experiment, the decapitated female was still preferentially courted in the choice assay (Fig. 2e), suggesting that female-specific chemicals are sufficient for sex-discrimination. As the exposure of both sexes to ozone impeded sex-discrimination, also the female cues seem to be sensitive to degradation by ozone. Indeed, when analyzing ozone-exposed females (i.e., 2 h at 100 ppb), we found reduced amounts of the described female-specific compounds 7,11-Heptacosadiene (7,11-HD) and 7,11-Nonacosadiene (7,11-ND)[22,23] (Fig. S7).

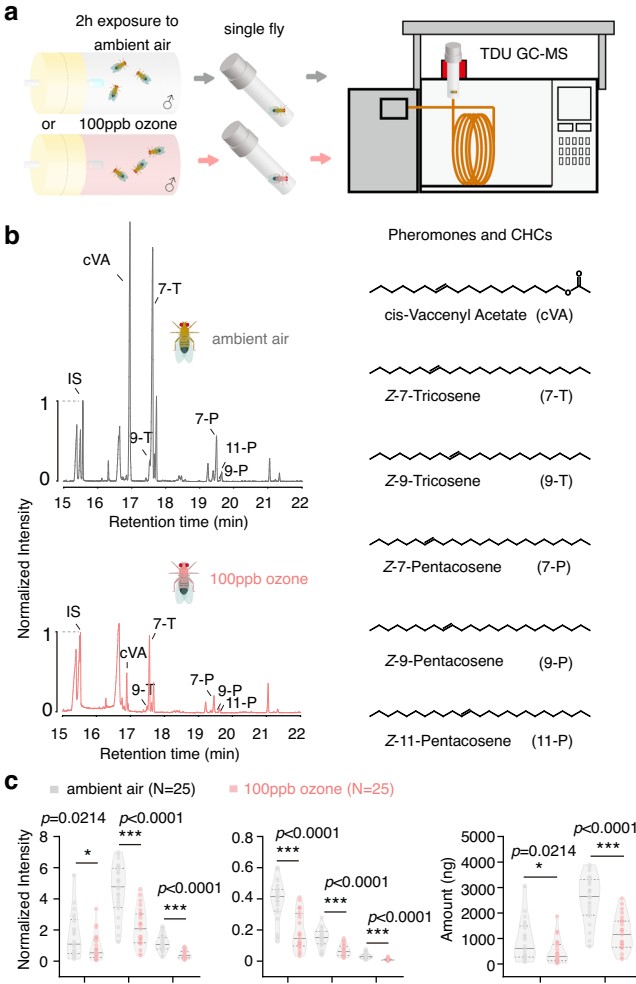

**Fig. 1 | TDU GC-MS analysis of *D. melanogaster* (CS) male pheromones and cuticular hydrocarbons. a** Schematic drawing of the TDU GC-MS analytical protocol. Male flies were first exposed to ozone or ambient air. Their chemical profile was then analyzed by TDU GC-MS. **b** Chemical profiles of *D. melanogaster* males after exposure to ozone (pink) or ambient air (grey). Chromatograms (left panel), Chemical structures of male pheromones and cuticular hydrocarbons, CHCs (right panel). **c**, Quantitative analysis of male pheromones and CHCs. (Two-tailed unpaired *t*-test; *p < 0.05; **p < 0.01; ***p < 0.001). Source data are provided as a Source Data file.

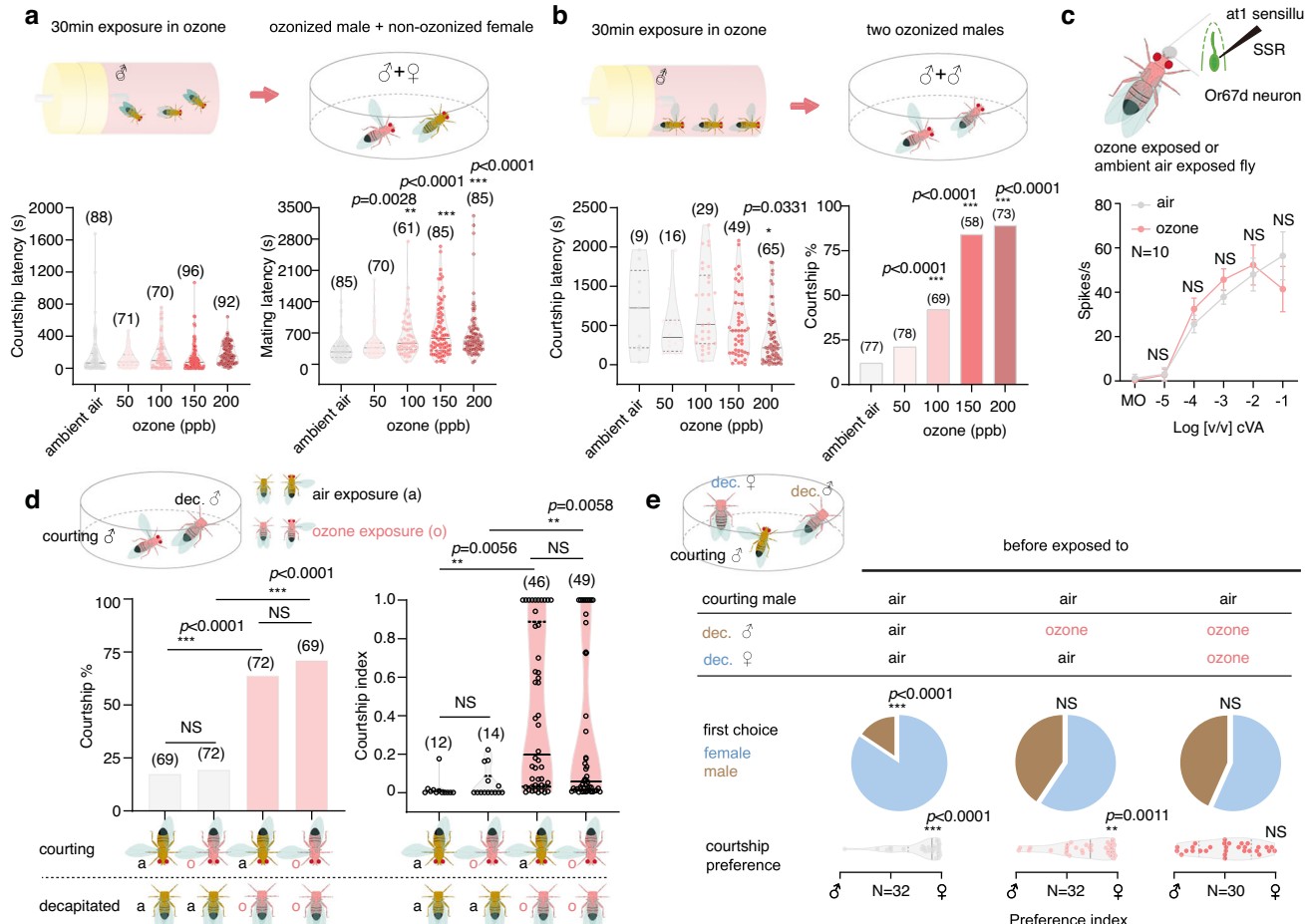

**Fig. 2 | Ozone exposure disrupts *D. melanogaster* (CS)'s sex discrimination.** **a** Male/female courtship latency and mating latency after male exposure to different levels of ozone (sample size provided in brackets, *Dunnett's* test for multiple comparisons against the ambient air control; groups significantly differing from control: *$p < 0.05$; **$p < 0.01$; ***$p < 0.001$). **b** Male/male courtship behavior after both males had been exposed to ozone. Courtship latency, *Dunnett's* test for multiple comparisons; courtship percentage (percentage of experiments that resulted in courting males), *Fisher's exact* test with *Holm-Bonferroni* correction for multiple comparison with control group; significant differences to control depicted as above. **c** SSR dose response curve of Or67d neuron (at1 sensillum) from male *D. melanogaster*. Top panel, schematic of SSR procedure; data presented as mean ± SE, $N = 10$ for either air or ozone exposure (*t*-test for each concentration; NS

indicates no significant difference). Mineral oil (MO) served as the solvent control. **d** Courtship behavior of an intact *D. melanogaster* male towards a decapitated male. Either the intact, the decapitated or both males were exposed to ozone before the courtship assay. Courtship percentage (see above), *Fisher's exact* test with *Holm-Bonferroni* correction for multiple comparison; courtship index (amount of time the intact male courted during the experiment), *Tukey's* test for multiple comparisons. **e** A male fly's preference for a decapitated male or female. Either the decapitated male, the female or both decapitated flies were exposed to ozone before the courtship assay. First choice presented in pie charts (*Fisher's exact* test); Preference index *(*time courting female – time courting male) / total courting time, unpaired *t*-test for courtship preference. All tests are two-tailed. Source data are provided as a Source Data file.

Having shown that exposure to ozone does not only reduce the attractiveness of a *D. melanogaster* (CS) male towards females but also strongly compromises sex discrimination, we asked whether this effect is specific to *D. melanogaster* (CS) or appears in other species as well. Based on our previous study on male-specific compounds in 99 *Drosophila* species[11], we selected 8 further *Drosophila* species that are known to carry male-specific compounds. We exposed their males for 2 h to 100 ppb ozone and compared their chemical profiles and behavioral performances with that of control males. Seven of these species, except *D. buskii*, exhibited decreased amounts of male-specific compounds after ozone exposure (Fig. 3). At the same time all of them showed decreased mating success and/or changes in male-male interactions (Fig. 3). *D. busckii*'s described male-specific compounds do not contain carbon-carbon double bonds and are thus less sensitive to degradation by ozone (Fig. 3). Although degrading effects of ozone on fly cuticular hydrocarbons lacking carbon double bonds have also been shown, those experiments were performed with ozone concentrations that with 45,000 ppm were around 10⁶ higher than those used in our experiments[24]. The fact that

*D. buskii* males were still less successful in courting females might be due to additional, non-identified but ozone-sensitive compounds governing courtship behavior in this species. Interestingly, both tested *D. mojavensis* subspecies tested, contrary to the other species, exhibited decreased male-male courtship after ozone-exposure. Again additional compounds to those described for these subspecies[10,11] might explain these results. We finally tested *D. suzukii*, a species that does not exhibit any sex-specific compounds and whose behavior is supposed to be rather visually driven[11,25]. As expected, neither mating success nor male-male courtship were affected by ozone treatment (Fig. 3).

Pollutants like ozone and nitric oxides have been shown to degrade floral volatiles and, hence, corrupt the chemical communication between plants and their pollinating insects[26–32]. Here we show that ozone in addition can harm insects in another context. The exposure of flies to rather mild ozone levels that have already been reported for polluted areas degrades their unsaturated pheromones and by that affects the sexual communication in 9 out of 10 tested *Drosophila* species. Carbon double bonds, however, are not specific

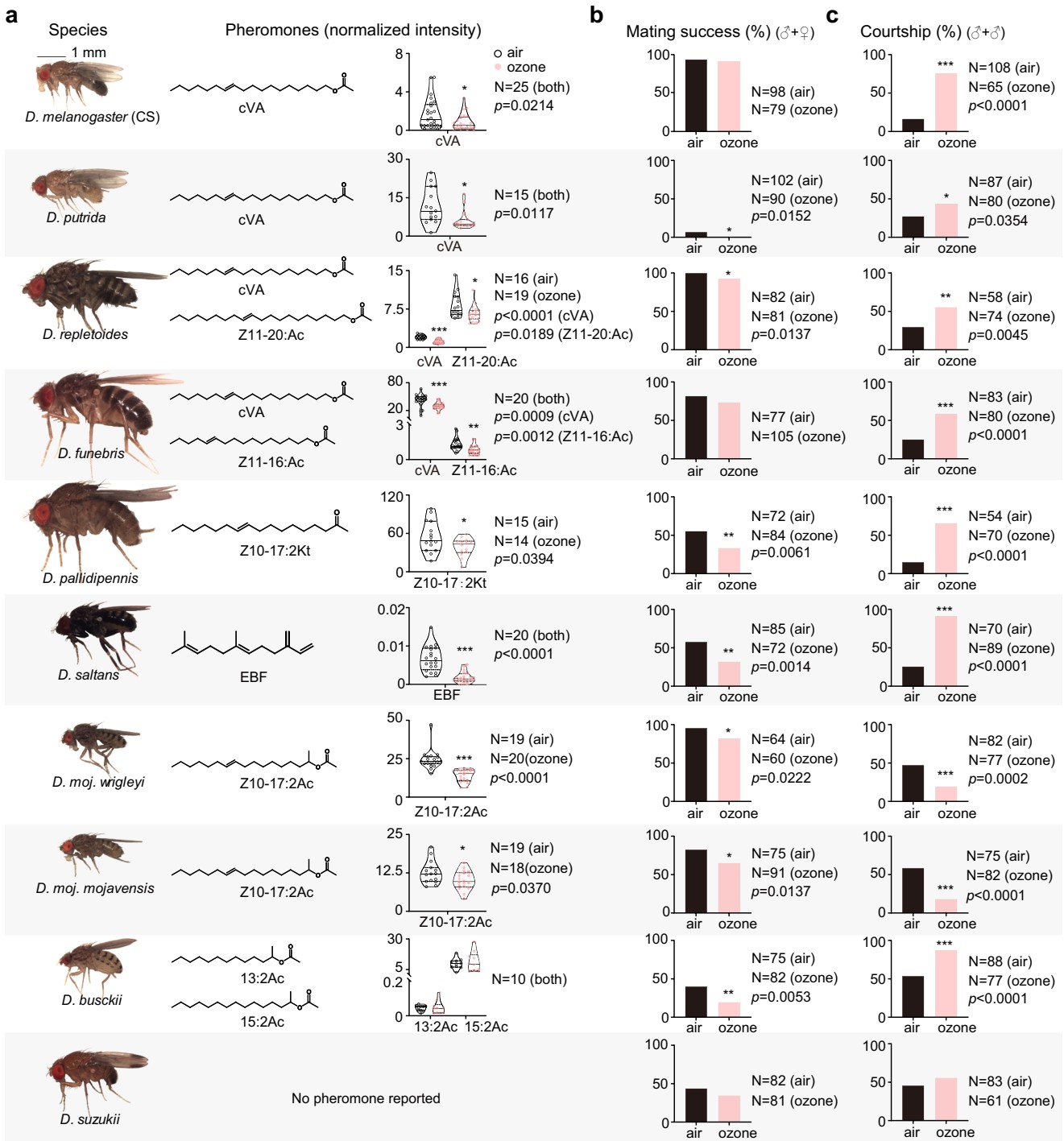

**Fig. 3 | Effects of ozone on male-specific compounds and sexual behavior of 10 drosophilid species. a** Data depict normalized peak areas of pheromones in ozone exposed and control flies (Two-tailed unpaired *t*-test; *$p < 0.05$; **$p < 0.01$; ***$p < 0.001$). **b**, **c** Effect on mating success (i.e., percentage of experiments resulting in mating) and effect on male-male courtship behavior (i.e., percentage of experiments resulting in male-male courtship). Two-tailed *Fisher's* exact test, *$p < 0.05$; **$p < 0.01$; ***$p < 0.001$. Source data are provided as a Source Data file.

for *Drosophila* pheromones but are a major characteristic of most identified insect pheromones[33]. As e.g., many female lepidopterans use pheromones for attracting males over long distances, there is ample time for the oxidizing pollutants to potentially degrade the signal before it reaches the receiver. While nowadays the detrimental effect of pesticides on insect populations is well established worldwide[34,35], insects obviously face a second problem: the degradation of their chemical information channels by increased levels of oxidizing pollutants.

## Methods

### *Drosophila* lines and chemicals stocks

Wild-type flies used in this study were obtained from the Bloomington Drosophila Stock Center (BDSC; https://bdsc.indiana.edu/index.html), National Drosophila Species Stock Center (NDSSC; http://blogs.cornell.edu/drosophila/), and Kyoto stock center (Kyoto DGGR; https://kyotofly.kit.jp/cgi-bin/stocks/index.cgi). The stock numbers are as following: *D. putrida* (15081–1401.00 / 15100–1711.01), *D. repletoides* (E-17001), *D. funebris* (15120–1911.05), *D. pallidipennis*

(15210–2331.02), *D. saltans* (14028–0571.00 / 15250–2451.01), *D. moj. wrigleyi* (15081–1352.22), *D. moj. mojavensis* (15081–1352.47), *D. busckii* (1300–0081.00), *D. suzukii* (14029–0011.01 / 14011–0131.04). For experiments with *D. melanogaster* we used the wild type strain Canton-S (CS). All flies were reared at 25 °C, 12 h light:12 h dark, and 70% relative humidity. Except for *D. melanogaster* (CS), all other fly species were bred with standard food mixed with banana. Virgins were collected by using $CO_2$ anesthesia. 10-day-old virgin flies were tested in the courtship arena. Care and treatment of all flies complied with all relevant ethical regulations.

## Chemicals

We synthesised fly compounds including (*Z*)–10-Heptadecen-2-one (Z10-17:2Kt), (*Z*)–11-Hexadecen-1-yl acetate (Z11-16:Ac), *rac*−2-Tridecyl acetate (13:2Ac), *rac*−2-Pentadecyl acetate (15:2Ac), (*Z*)–11-Eicosen-1-yl acetate (Z11-20:Ac), (*Z*)–10-Heptadecen-2-yl acetate (Z10-17:2Ac) as described[11]. Other chemicals including cis vanccenyl acetate (cVA), (*Z*)−7-Tricosene (7-T), (*Z*)−9-Tricosene (9-T), (*Z*)−7-Pentacosene (7-P), and (*Z*)−9-Pentacosene (9-P), were purchased in high purity from Sigma-Aldrich and Cayman Chemical.

**(Z)−11-Pentacosene synthesis.** (*Z*)−11-Pentacosene (11-P) was synthesized via Wittig reaction from bromotetradecane (Sigma-Adrich, Germany) and undecanal (Sigma-Aldrich, Germany). Bromotetradecane (1 g, 3.6 mmol) and triphenylphosphine (946 mg, 3.6 mmol) were dissolved in 20 ml of Toluene and heated at reflux for 20 h. After cooling to room temperature, the toluene phase was carefully removed with a pipette. The residue was stirred for 5 min with fresh toluene (10 ml each), before the toluene was pipetted off again. This process was repeated three times with toluene and once with diethylether (10 ml). The highly viscous residue was dried in high vacuum overnight to obtain the Wittig-salt as an amorphous off white solid.

Under argon the Wittig-salt (600 mg, 1.11 mmol) was dissolved/suspended in anhydrous THF (20 ml) and cooled to −30 °C. A solution of sodium bis(trimethylsilyl)amide (1.2 ml, 1 M in THF) was added dropwise and the mixture was allowed to warm to −10 °C. After stirring for 30 min the now deep orange suspension was cooled to −30 °C again before adding undecanal (0.23 mL, 1.11 mmol) via syringe. The mixture was stirred overnight at 20 °C. The reaction mixture was diluted with n-hexane (20 ml) before adding water (40 ml). The organic phase was removed and the aqueous phase was extracted with n-hexane (2 × 40 ml). The combined organic phases were washed with water and brine (40 ml each) and dried over $Mg_2SO_4$. After filtration the solvent was removed via rotavap, the residue adsorbed onto silica gel and chromatographed with n-hexan/EtOAc (50:1) as eluent to obtain (*Z*)−11-pentacosene (130 mg, 34% yield based on undecanal) as a colorless oil.

## Ozone exposure system

Ozone exposure system see Fig. S1. Compressed ambient air (with 4.5 ± 0.5 ppb ozone) was used to produce control air (i.e., ambient air humidified to 70% relative humidity), clean air (i.e., air that via a palladium ozone scrubber was cleaned from all naturally occurring ozone) and ozone enriched air (i.e., ambient air that was led through an ozone generator (Aqua Medic, Germany) which could produce up to 100 mg ozone/h). By humidifying and mixing clean with ozone-enriched air in the mixing box, different levels of ozonated experimental air could be produced. Ozone concentration could be increased (decreased) by decreasing (increasing) the flow rate through the mass flow controller before the ozone scrubber (e.g., 5 l/min), while the flow rate through the ozone generator was kept constant at 0.7 l/min. As the ozone concentrations changes when air becomes humidified, ozonated experimental air was dynamically stored in a mix box (a 100 L Plexiglas container), from which air was continuously probed for the ozone monitor (BMT 932, BMT Messtechnik GmbH,

Germany), while at the same time, each 0.2 l/min were led into the four 70 ml plastic vials containing the flies, and surplus air was cleaned from ozone via an additional palladium ozone scrubber before releasing it into the chamber air. A second set of four 70 ml vials was connected to the airflow of ambient air to produce "control flies" for the experiment. The whole system was built twice with a valve switching the ozone-enriched air between the two setups so that another set of experimental and control flies could be prepared in parallel. By e.g., opening the valve 5 s towards the left and 10 s towards the right setup, experiments with different ozone concentrations could be run in parallel.

## TDU GC-MS

To analyze chemical profiles of flies in the TDU GC-MS experiments, 10 days old flies were first exposed to ozone or control air for a given time and afterwards immediately frozen at −20 °C for 30 min. In order to investigate the recovery of chemical profiles after ozone exposure, males were exposed to 100ppb ozone for 2 h, and then moved to tubes with standard food for either 1 or 5 days. Afterwards they were frozen as mentioned before. For TDU GC-MS measurements, individual flies were placed in microvials of thermal desorption tubes (GERSTEL, Germany) and 0.5 μl of C10-Br or C16-Br ($10^{-3}$ dilution in hexane) were added to the microvials as internal standard.

Desorption tubes were transferred using a GERSTEL MPS 2 XL multipurpose sampler into a GERSTEL thermal desorption unit (GERSTEL, Germany). Samples were desorbed at 250 °C for 8 min, then trapped at −50 °C in the liner of a GERSTEL CIS 4 Cooled Injection System, with liquid nitrogen used for cooling. The components were transferred to the GC column by heating the programmable temperature vaporizer injector at 12 °C/s up to 270 °C and then keeping the temperature for 5 min. The GC-MS (Agilent GC 7890A fitted with an MS 5975 C inert XL MSD unit; Agilent Technologies, USA) was equipped with an HP5 column (Agilent Technologies, USA). The temperature of the gas chromatograph oven was held at 50 °C for 3 min and then increased by 15 °C /min to 230 °C and then by 20 °C /min to 280 °C, held for 20 min. Mass spectra were taken in EI mode (at 70 eV) in the range from 33 m/z to 500 m/z.

## Courtship behavior assays

Virgin males and females of all tested drosophilids were collected after eclosion and raised individually and in groups (20 individuals/vial), respectively. For male-female courtship assays, 15 males were exposed to ozone or ambient air in a vial. An ozone-treated individual male was then placed with an untreated female into a courtship chamber and their behavior was observed and quantified for 1 h. Each courtship arena contained 4 chambers (1 cm diameter × 0.5 cm depth) covered with a plastic slide. Four chambers were recorded simultaneously. Air flow of 0.2 mL/min was added from below to each arena. A GoPro Camera 4 or Logitech C615 was used to record courtship behaviors. Each video was analyzed manually for courtship latency (i.e., the time until the male initiated courtship behavior), courtship percentage (i.e., the percentage of males that showed courtship behavior), courtship index (i.e., the time each male performed courtship behavior during the 10 min experimental time), mating latency (i.e., the time until successful mating started), and mating success (i.e., the percentage of males that copulated). All behavioral experiments were performed at 25 °C and 70% humidity.

For male-male courtship assays, 25 males were exposed to ozone or ambient air in a fly tube. Afterwards, two males were put into one chamber and their behaviors observed and quantified for 30 min. For *D. melanogaster* (CS), several ozone exposure combinations (i.e., 15 min, 30 min, and 2 h with 50, 100, 150, 200 ppb, respectively) were tested. Males of other drosophilids were exposed for 2 h to 100 ppb. For no-choice assays with decapitated males, males were exposed to ozone, and then decapitated. An intact male and a decapitated male

were put into one chamber and the courtship behavior of the intact male observed and quantified for 30 min. For the two-choice assays with decapitated flies, males and/or females were exposed to ozone, and then decapitated. An intact male, and a decapitated male and female were put into one chamber. The preference index of the courting intact male was calculated as (time taken for an intact male to court a decapitated female−the time taken for an intact male to court a decapitated male)/30 min.

### Single sensillum recording (SSR)

Male *D. melanogaster* (CS) flies were exposed to either ozone or ambient air, and then immobilized in a pipette tip. A reference electrode was put into the eye; another tungsten electrode was inserted into the target sensillum. The at1 was identified based on their location and spontaneous activities. Signals were amplified by Syntech Universal AC/DC Probe (Syntech, Germany), sampled (96,000.0 samples/s), and filtered (500–5000 Hz with 50/ 60 Hz suppression) via a USB-IDAC (Syntech, Germany) connection to a computer. Action potentials were extracted using AutoSpike software (Syntech, Germany). Synthetic compounds were diluted in mineral oil (MO) (Sigma-Aldrich, Germany). Before the test, 10 µl of the diluted odor was freshly loaded onto a small piece of filter paper ($1 \, cm^2$), and placed inside a glass Pasteur pipette. The tested odor dosages were ranging from $10^{-5}$–$10^{-1}$ dilution (v/v). The odorant was delivered by inserting the tip of the pipette into a constant, humidified airstream flowing at 600 ml/min through a stainless steel tube (diameter, 8 mm) ending in 1 cm distance from the antenna. Neural activity was recorded for 10 s, starting 3 s before the stimulation period of 0.5 s. Responses from individual neurons were calculated as the increase (or decrease) in the action potential frequency (spikes/s) relative to the pre-stimulus frequency. Traces were processed by sorting spike amplitudes in AutoSpike, analysis in Excel, and illustration in Adobe Illustrator CS (Adobe systems, USA).

### Statistical analyses

Statistical analyses (see the corresponding legends of each figure) and preliminary figures were conducted using GraphPad Prism v. 8 (GraphPad Software, USA). Figures were then processed with Adobe Illustrator CS5.

### Reporting summary

Further information on research design is available in the Nature Portfolio Reporting Summary linked to this article.

## Data availability

All the data generated in this study are provided in the Source Data file. Source data are provided with this paper.

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

## Acknowledgements
We thank I. Alali and S. Trautheim for help with the fly breeding. This research was supported through funding by the Max Planck Society and specifically through funding to the Max Planck Center "Next Generation Insect Chemical Ecology.".

## Author contributions
NJ.J., B.S.H., and M.K. designed the research plan and NJ.J. performed most of the experiments. H.C. conducted SSR experiments. F.E. and D.V. constructed the ozone exposure device. NJ.J and K.W. analyzed and quantified pheromone compounds. J.W. synthesized synthetic compounds. NJ.J. analyzed experimental data. NJ.J., B. S.H., and M.K. wrote the paper. All authors edited the manuscript.

## Funding

## Competing interests
The authors declare no competing interests.
