## [Peer Review File · Nature Communications]

Ozone exposure disrupts insect sexual communicationREVIEWER COMMENTS

Reviewer #1 (Remarks to the Author):

Review of 'Increased ozone levels disrupt insect sexual communication', by Jiang et al. This is an interesting study of the effects of exposure to elevated ozone on the sexual communication of a range of drosophilids, with particular focus on *Drosophila melanogaster*. It is clearly shown that ozone has a strong effect on sexual recognition, with males rendered slower to mate with females and less able to distinguish males from females. It is an interesting observation that is presented clearly and with some compelling video documentation in the supplementary material. The work represents a very interesting observational and mechanistic study suggesting that oxidizing air pollutants could have substantial effects on pheromonal communication. It is further clarified that exposure to ozone does not significantly affect the ability of the tested insects to detect the pheromone, which provides credence to the mechanism most likely being the degradation of pheromone. The research presented is mostly clearly explained, although I have a few relatively minor queries concerning the methodology. My assessment is that the conclusions are justified as they are presented and the methods appear suitably robust for a study of this type. The question that is begged by this study is whether these strong effects of air pollutants on sexual communication occur under real world conditions when pollution levels are high. Future research should perhaps look at the pheromonal composition of flies collected from areas of different pollution levels.

Comments on the materials and methods:

Lines 302-305: There is not much information about the different *Drosophila* species used. Only the one strain of *D. melanogaster*, which is not actually defined as *D. melanogaster*. More information is needed.

Lines 313-328: If there is a restriction on space for the methods, this extensive section on synthesis of (Z)-11-Pentacosene could be moved to supplementary material in favor of including details more pertinent to the study.

Lines 331-336: I find the description of the ozone exposure system too brief. There is a nice figure, Fig S1 (not S8 as stated on line 336), that represents the different components of the system, but the text does not include anything about the control air, the timing of the valve system for switching between samples, the presence of two mixing boxes, nor the volume of the tubes containing the flies. The flow rates between the different parts of the system could also be described. In fact, aside from the 100L Plexiglas chamber, which is presumably labelled Mix box in fig S1, there is no information on the volume of any of the mixing vessels within the system. I recommend that these details are added and that the description of the set-up is in general given more depth.

Lines 358-359: Clarify if individual males were placed with an untreated female. The text is not clear on that. Did the set up enable the recording of four courtship chambers simultaneously? Also, it is unclear what the '(see below)' refers to.

Lines 365: It should be clarified how the percentage of males that successfully mated was determined. Was this just an observation of a mating event, or viable reproduction?

Lines 375: I guess that this calculation is the 'time taken for an intact male to court a decapitated female – the time taken for an intact male to court a decapitated male'. If so, the phrasing is currently a bit strange.

Other minor comments:

Lines 51-52: I suggest to include some newer references for the extreme ozone events and to indicate how long the episodes tend to last and perhaps the regions/areas being referred to.

Line 107, line 138: I suggest not to start these sentences with 'obviously'.

Line 123: The 'fruitless' mutation may mean little to many readers without a few words explaining its significance.

Line 143-144: I suggest rephrasing this sentence.

Line 169-170: Again, I suggest rephrasing.

Line 175: Rephrase to ...carbon bonds has also been shown...

In figure 3: The grey bars on a grey background can be a bit unclear. I would suggest having colors with a stronger contrast. Likewise, in figure 2, part d, I found the colors of the flies

indicating the treatments a bit unclear. It was possible to see the colors well when increasing the size of the figure, but at normal size it was a bit obscure.

Reviewer #2 (Remarks to the Author):

The manuscript by Jiang et al. presents very interesting and complete data on the influence of increased ozone levels on pheromone communication in *Drosophila melanogaster* and several other fruitfly species. This is an important and highly original work in view of the process of climate change and anthropogenic changes in the environment. The elegantly designed multidisciplinary experiments on a model organism show convincingly that the effect of ozone is due to degradation of carbon double bonds in the pheromone, whereas no influence on pheromone reception was shown. The comparative approach, investigating different *Drosophila* species, is an important step to generalize the findings on the model organism. Only species with pheromone components with double carbon bonds seem to be affected by ozone, whereas behavior in *Drosophila suzukii*, a species with dominant visual courting behavior, seems not to be affected. The paper presents a large amount of highly condensed data, which are excellently presented in complex, but comprehensive Figures. Materials and Methods are described in sufficient detail to reproduce experiments. Important additional information is presented in supplementary figures and movies. In addition, the broader impact of the findings is very well discussed (in the context of reproductive success in both pest and "beneficial" insects under increased ozone levels), and opens up highly interesting perspectives for future research in this field.

I only found a few small errors and missing details, which should be corrected:

Fig 1 legend: cuticular hydrocarbons (not curricular)

Fig. S4 legend: please mention the sex of the exposed flies (males?)

Fig. S5 legend: please mention the time of exposure (2 h according to text) and the ozone concentration used in this case

Page 8 line 11: ... carbon double bonds have been shown....

Sylvia Anton

RESPONSE TO REVIEWERS' COMMENTS

Reviewer #1 (Remarks to the Author):

Review of 'Increased ozone levels disrupt insect sexual communication', by Jiang et al.

This is an interesting study of the effects of exposure to elevated ozone on the sexual communication of a range of drosophilids, with particular focus on *Drosophila melanogaster*. It is clearly shown that ozone has a strong effect on sexual recognition, with males rendered slower to mate with females and less able to distinguish males from females. It is an interesting observation that is presented clearly and with some compelling video documentation in the supplementary material. The work represents a very interesting observational and mechanistic study suggesting that oxidizing air pollutants could have substantial effects on pheromonal communication. It is further clarified that exposure to ozone does not significantly affect the ability of the tested insects to detect the pheromone, which provides credence to the mechanism most likely being the degradation of pheromone. The research presented is mostly clearly explained, although I have a few relatively minor queries concerning the methodology. My assessment is that the conclusions are justified as they are presented and the methods appear suitably robust for a study of this type. The question that is begged by this study is whether these strong effects of air pollutants on sexual communication occur under real world conditions when pollution levels are high. Future research should perhaps look at the pheromonal composition of flies collected from areas of different pollution levels.

Response to the Review Comments:

We would like to thank you for your helpful comments and constructive suggestions, which have significantly improved our manuscript. We fully agree, that comparing flies from differentially polluted areas will be very interesting. Such a transect is actually part of a grant application that we are submitting soon.

Comments on the materials and methods:

Lines 302-305: There is not much information about the different *Drosophila* species used. Only the one strain of *D. melanogaster*, which is not actually defined as *D. melanogaster*. More information is needed.

R: Thank you for comments. We have added more information about stock numbers and information on fly breeding to the materials and methods part. See lines 316-324. We change our description of *D. melanogaster* to *D. melanogaster* (CS).

Lines 313-328: If there is a restriction on space for the methods, this extensive section on synthesis of (Z)-11-Pentacosene could be moved to supplementary material in favor of including details more pertinent to the study.

R: Thank you for your suggestions, we have moved the synthesis of (Z)-11-Pentacosene to the supplementary method part. See lines 91-101.

Lines 331-336: I find the description of the ozone exposure system too brief. There is a nice figure, Fig S1 (not S8 as stated on line 336), that represents the different components of the system, but the text does not include anything about the control air, the timing of the valve system for switching between samples, the presence of two mixing boxes, nor the volume of the tubes containing the flies. The flow rates between the different parts of the system could also be described. In fact, aside from the 100L Plexiglas chamber, which is presumably labelled Mix box in fig S1, there is no information on the volume of any of the mixing vessels within the system. I recommend that these details are added and that the description of the set-up is in general given more depth.

R: Thank you for your comments and for picking the wrong figure reference in the manuscript. We have corrected the figure number in the text and have added a detailed description of the ozone exposure system lines 335-351. While doing so, we also realized that some parts of the figure were a bit unclear. We therefore, revised the Fig S1 and hope that adding the new methods part and revising the figure circumvents further confusion.

Lines 358-359: Clarify if individual males were placed with an untreated female. The text is not clear on that. Did the set up enable the recording of four courtship chambers simultaneously? Also, it is unclear what the '(see below)' refers to.

R: Thank you. We have reformulated this sentence to "An ozone-treated individual male was then placed with an untreated female into a courtship chamber and their behavior was observed and quantified for 1h. "in lines 373-375. We also added "Four chambers were recorded simultaneously" in lines 375. We have also deleted "see below".

Lines 365: It should be clarified how the percentage of males that successfully mated was determined. Was this just an observation of a mating event, or viable reproduction?

R: We have rewritten the sentence to "and mating success (i.e. the percentage of males that copulated)" in lines 380.

Lines 375: I guess that this calculation is the 'time taken for an intact male to court a decapitated female – the time taken for an intact male to court a decapitated male'. If so, the phrasing is currently a bit strange.

R: Thank you for suggestion, we now use your suggested sentence instead in line 389-390.

Other minor comments:

Lines 51-52: I suggest to include some newer references for the extreme ozone events and to indicate how long the episodes tend to last and perhaps the regions/areas being referred to.

R: We agree that this information is useful and have added “Local extreme ozone events have been reported for industrial and urban areas of e.g. Mexico, Bangladesh, Morocco, and China¹⁷⁻²⁰. Ozone reached in Mexico a concentration of up to 210 ppb (that lasted an hour) and the highest mean value measured over 10 hours exceeded 170 ppb¹⁷. Although ozone values vary a lot during the whole year, the yearly measured average for e.g. north-eastern China has increased from 45 ppb in 2003 to 62 ppb in 2015¹⁸ and reached a daily maximum averaged for 8 hours (MDA8) of 140 ppb observed in March 2020²¹.” to the introduction. (lines 51-57.)

Line 107, line 138: I suggest not to start these sentences with ‘obviously’.

R: We have exchanged “obviously” in lines 114 and line 146.

Line 123: The ‘fruitless’ mutation may mean little to many readers without a few words explaining its significance.

R: We apologize for not explaining this term and have now added “formed a long chain of courting males that was first described for males carrying a *fruitless* mutation, i.e. a mutation in the *fruitless* gene that changes the males’ mating preferences” to lines 129-131.

Line 143-144: I suggest rephrasing this sentence.

R: Thank you. We rephrased the sentence to “Before the test, the decapitated flies were exposed to either ozone or control air.” in lines 151-152.

Line 169-170: Again, I suggest rephrasing.

R: We rephrased the sentence to “Seven of these species, except *D. buskii*, exhibited decreased amounts of male-specific compounds after ozone exposure (Fig. 3). At the same time all of them showed decreased mating success and/or changes in male-male interactions (Fig. 3).” in lines 171-180.

Line 175: Rephrase to ...carbon bonds has also been shown...

R: Done in line 183.

In figure 3: The grey bars on a grey background can be a bit unclear. I would suggest having colors with a stronger contrast. Likewise, in figure 2, part d, I found the colors of the flies indicating the treatments a bit unclear. It was possible to see the colors well when increasing the size of the figure, but at normal size it was a bit obscure.

R: Thank you for comments. We adjusted both Figures accordingly.

Reviewer #2 (Remarks to the Author):

The manuscript by Jiang et al. presents very interesting and complete data on the influence of increased ozone levels on pheromone communication in *Drosophila melanogaster* and several other fruitfly species. This is an important and highly original work in view of the process of climate change and anthropogenic changes in the environment. The elegantly designed multidisciplinary experiments on a model organism show convincingly that the effect of ozone is due to degradation of carbon double bonds in the pheromone, whereas no influence on pheromone reception was shown. The comparative approach, investigating different *Drosophila* species, is an important step to generalize the findings on the model organism. Only species with pheromone components with double carbon bonds seem to be affected by ozone, whereas behavior in *Drosophila suzukii*, a species with dominant visual courting behavior, seems not to be affected. The paper presents a large amount of highly condensed data, which are excellently presented in complex, but comprehensive Figures. Materials and Methods are described in sufficient detail to reproduce experiments. Important additional information is presented in supplementary figures and movies. In addition, the broader impact of the findings is very well discussed (in the context of reproductive success in both pest and "beneficial" insects under increased ozone levels), and opens up highly interesting perspectives for future research in this field.

Response to the Review Comments:

We thank the reviewer for the positive opinion and the very constructive suggestions.

I only found a few small errors and missing details, which should be corrected:

Fig 1 legend: cuticular hydrocarbons (not curricular)

R: Thank you. We change in line 78.

Fig. S4 legend: please mention the sex of the exposed flies (males?)

R: Thank you for the suggestion. We add in line 46-47.

Fig. S5 legend: please mention the time of exposure (2 h according to text) and the ozone concentration used in this case

R: We have added more information about time of exposure and ozone concentration to the legend of Fig S5 in line 56-57.

Page 8 line 11: ... carbon double bonds have been shown....

R: Thank you for the comment. We rephrased in line 183.

REVIEWERS' COMMENTS

Reviewer #1 (Remarks to the Author):

All of my earlier comments have been satisfactorily addressed. Congratulations on an interesting study and good luck with the follow up research.